



# Wave forecast investigations on downscaling, source terms, and tides for Aotearoa New Zealand

Rafael Santana[1,2], Richard Gorman[3], Emily Lane[4], Stuart Moore[5], Cyprien Bosserelle[4], Glen Reeve[1], and Christo Rautenbach[1,6]

[1]National Institute of Water and Atmospheric Research, Coasts and Estuaries, Hamilton, Aotearoa New Zealand
[2]Department of Physics, The University of Auckland, Auckland, 1010, Aotearoa New Zealand
[3]Spectrum Oceanographic Ltd, Picton, Aotearoa New Zealand
[4]National Institute of Water and Atmospheric Research, Hydrodynamics, Christchurch, Aotearoa New Zealand
[5]National Institute of Water and Atmospheric Research, Meteorology and Remote Sensing, Wellington, Aotearoa New Zealand
[6]Institute for Coastal and Marine Research, Nelson Mandela University (NMU), Port Elizabeth, South Africa

**Correspondence:** Rafael Santana (Rafael.Santana@niwa.co.nz)

**Abstract.**

This study evaluates the effects of downscaling, source terms, and tidal interactions on numerical wave forecasts in Aotearoa New Zealand. We utilised a set of three nested domains (from global to regional scale) to examine significant wave height (Hs), mean period (Tm01), and peak wave direction at two coastal locations, Banks Peninsula and Baring Head. Downscaling

markedly improved forecast accuracy at Baring Head, a tidally constricted region, reducing Hs forecast error by 25%. However, improvements at Banks Peninsula were minimal, likely due to its open coast characteristics which are adequately represented even by lower resolution models. Source term enhancements using default ST6 parameters generally improved Hs predictions on the west coast but worsened them on the east, indicating a geographical dependency in model performance. This variability was also evident in the Tm01 predictions, with notable improvements in bias reduction through model downscaling, particularly

at Baring Head. Tidal influences were significant, especially at Baring Head, where they enhanced the forecast accuracy of wave height and direction due to the strong tidal currents characteristic of this location. In contrast, at Banks Peninsula, tidal effects were less pronounced. The study underscores the importance of tailored modelling approaches that consider local geographical and hydrodynamic conditions to optimise wave forecasting.

## 1 Introduction

Understanding local wave variability is an important aspect of most coastal engineering projects (e.g., Camus et al., 2011; Kamphuis, 2020; Kroon et al., 2020). Beyond these studies, there are numerous ecological (e.g., Coppin et al., 2020), infrastructure and logistics (e.g., Camus et al., 2019; Lucio et al., 2024), and coastal safety (e.g., de Vos and Rautenbach, 2019; Altomare et al., 2020) questions that require a thorough description of offshore and coastal waves. Wave climate studies are usually derived from longterm hindcasts or reanalyses. Wave variability can also be described in context of extreme (Simmonds and Keay, 2000) and ambient wave climates (Mortlock and Goodwin, 2015). Forecasting ocean waves is important





for oceanic traffic safety, on- and offshore industrial operations and recreational activities. Extreme event forecasting (and the associated accuracy in magnitude and timing) has obvious implications for coastal safety and infrastructure. Increased sea level and cyclone activity due to climate change (e.g., Vitousek et al., 2017; Diamond and Renwick, 2015) has implications for infrastructure design, coastal adaptation and conservation (Toimil et al., 2020). Examples of these include coastal areas having

increased susceptibility to wave impact due to rising sea levels (Hannah, 2004; Hannah and Bell, 2012; Hauer et al., 2016) and the potential of increased storminess (Albuquerque et al., 2024).

Currently, several global wave forecasts are freely available, e.g., the Global Forecast System by the National Centers for Environmental Prediction (NCEP) (Tolman et al., 2002), ERA5 by the European Centre for Medium-Range Weather Forecasts (ECMWF) (Hersbach et al., 2018), and WAVERYS by Copernicus Marine Environment Monitoring Service (CMEMS) (Law-

Chune et al., 2021). However, these wave forecast systems use low-resolution models ($> 20$ km) that cannot represent complex bathymetry and sometimes fail at correctly simulating large coastal wave events (Fanti et al., 2023). Therefore, regional and local wave forecasts are important.

In Aotearoa New Zealand, the National Institute of Water and Atmospheric Research (NIWA) is tasked to conduct research and develop tools aimed at enhancing the countries' resilience to wave-related and other environmental hazards. Being an

island nation, New Zealand has an extensive Exclusive Economic Zone (EEZ) and coastline to manage. NIWA has developed a platform called EcoConnect, that generates and disseminates tailored environmental information services in near-real time (Webster et al., 2008; Moore et al., 2022). This platform manages operational forecasts of atmospheric, hydrological, storm tide, and wave conditions. EcoConnect also disseminates outputs from numerical models along with data obtained from various observational installations.

EcoConnect forecasts waves routinely using Wave Watch III® (WW3) which is a third-generation spectral wave model that uses an energy-based approach to describe the physical processes of wave growth and transformation at oceanic scales (WW3DG, 2019). WW3 is implemented within EcoConnect using a set of three nested grids to appropriately resolve spatial scales in the wave field surrounding the country (refer to Fig. 1). An adequate model resolution is needed for different regions of Aotearoa New Zealand. Open ocean regions, such as the west and east coasts of the South Island, may be adequately simulated

using low-resolution models. Conversely, regions with complex coastlines, such as the Cook Strait or Hauraki Gulf, could require higher-resolution models ($<4$ km). In both cases validation is necessary to identify regions that need increased model resolution and vice-versa.

The WW3 models used in EcoConnect all use the same parameter settings derived from a calibration study (Gorman and Oliver, 2018) in which a global wave model was auto-calibrated against satellite altimetry data, using an iterative process

that finds values for each model parameter which minimises the root-mean-square difference between model and observed significant wave heights over the global model domain. Initial short-term calibration tests were conducted using two alternative input/dissipation source term packages, namely ST2 and ST4, in which ST4 performed better and was selected for a 1-year calibration. ST4 physics include developments in ST1 and ST3 and the latter source terms were not tested by Gorman and Oliver (2018). However, ST6 physics (Rogers et al., 2012; Zieger et al., 2015) hasn't been tested in EcoConnect's WW3

implementations yet. WW3 ST6 is similar to Simulating Waves in the Nearshore (SWAN, Holthuijsen et al., 1993; Ris et al.,







**Figure 1.** (a) Nested model domains and respective forecasted wave fields for 6 am UTC 29 of Jun 2021. Colour shade represents significant wave height from each domain. Global model outputs are shown as the most external colour shade. Basin scale model outputs are shown inside the largest black contour area. The innermost contour highlights the area of the high-resolution model around the mainland of Aotearoa New Zealand. The white arrows show wave height and peak wave direction and are plotted every 40th grid point. The red circles are the location of *in situ* measurements from wave buoys at Banks Peninsula and Baring Head (the northernmost red dot in this figure). The 200-m isobath is shown as a dark red contour.

1995) model physics, which is meant for coastal applications. ST6 models have shown improved/similar results compared to





ST4 experiments during storm passage and ambient conditions (e.g. Kalourazi et al., 2021; Zou et al., 2023; Meucci et al., 2023b) and has also been used for global wave climate simulations (e.g. Meucci et al., 2023a).

Ocean currents also create important variability in the wave field and it has been an active topic of research in the last decade (e.g. Zhang et al., 2022). Wave-current interaction studies have shown variability ranging from large-scale oceanic currents (e.g. Barnes and Rautenbach, 2020) to small-scale tidally-dominated regions (e.g., Vincent, 1979; Ris et al., 1995). For instance, a wave train propagating against opposing currents tend to increase wave steepness which in turn can generate wave breaking and dissipation (Holthuijsen, 2007). Opposing currents can also reduce the wave period/length and have been called "wave straining" by Holthuijsen and Tolman (1991). These effects have been observed and simulated in different regions of the globe using short-term timeseries (e.g., Ardhuin et al., 2012; Rapizo et al., 2017; Halsne et al., 2024). However, a longterm (>1 year) evaluation of the importance of tidal currents on the wave forecast is still needed.

The aim of the present study is to analyse the impact of wave model downscaling (including storm tide forcing) on wave forecast for Aotearoa New Zealand. The impact of downscaling is assessed using a set of three nested model domains with increasing resolution but with similar grid (i and j) dimensions. Thus implying the same computational cost. A comparison between source terms physics (ST4 and ST6) is made using the intermediate model grid. The impact of tides and storm surge forcing is analysed using the model with the highest resolution in which two experiments are compared: with and without tidal and storm surge forcing. Both *in situ* and remotely sensed validation/investigations are performed and the physical dynamics of New Zealand's wave climate are discussed, especially in the Cook Strait.

## 2   Methods

### 2.1   The operational forecasting system

The EcoConnect platform, comprising data ingestion, numerical modelling applications for a variety of natural hazards and forecast data delivery. EcoConnect operates autonomously via the Cylc workflow meta-scheduler (Oliver et al., 2018) and starts by downloading the United Kingdom Met Office's global model atmospheric forecast (UKMO). This model is a configuration of the Unified Model (UM, Maher and Earnshaw, 2022), and provides lateral boundary conditions for the New Zealand Limited Area Model (NZLAM) atmospheric model, itself a local configuration of the UM whose domain extends from Eastern Australia to the Chatham Islands (external black contour in Fig. 1). NZLAM provides the initial and lateral boundary conditions for the New Zealand Convective-Scale Model (NZCSM) numerical model, a convection-permitting configuration of the UM covering just the New Zealand's landmass and its coastal waters (innermost domain in Fig. 1).

Both NZLAM and NZCSM are configured to provide input data for suite of different hazards models. These include a hydro-logical river flow model, TopNet, forecasts streamflow for just under 50,000 river reaches around New Zealand (Cattoën et al., 2022) , and a hierarchy of wave and current forecast models, based on WW3 and the River and Coastal Ocean Model (RiCOM) (more details about these models can be found in sections 2.2 and 2.4). Observation datasets collected and disseminated within EcoConnect include satellite imagery, surface weather station, river gauges and wave buoys data. These model outputs are generated, processed, compiled and archived by bespoke tasks in the EcoConnect workflow, all orchestrated by Cylc.





## 2.2 Wave model


The WW3 version 6.07.1 (WW3DG, 2019) is used for wave forecasts in EcoConnect. The model represents the sea state by the two-dimensional ocean wave spectrum $F(\boldsymbol{k},\boldsymbol{x},t)$, which gives the energy density of the wave field as a function of wavenumber $\boldsymbol{k}=(k_x,k_y)$, at each position $\boldsymbol{x}=(x,y)$ in the model grid and time $t$ of the simulation. The spectrum evolves subject to a radiative transfer equation,

$$\frac{\partial N}{\partial t} + \boldsymbol{\nabla}_{(x,y)} \cdot (\dot{\boldsymbol{x}}N) + \frac{\partial}{\partial k}(\dot{k}N) + \frac{\partial}{\partial \theta}(\dot{\theta}N) = \frac{S}{\sigma}$$


(1)

for the wave action $N(k,\theta,\boldsymbol{x},t) = F(\boldsymbol{k},\boldsymbol{x},t)/\sigma(k)$, where the dots represent time derivatives, $\theta$ is the propagation direction and $\sigma=2\pi f$ is the relative (radian) frequency associated with waves of wavenumber magnitude $k$ through the linear dispersion relation

$$\sigma^2 = gk \tanh kd \tag{2}$$

The frequency $\sigma$ of waves propagating subject to gravity $g$ and water depth $d$, is observed relative to a frame of reference moving with a mean current $\boldsymbol{U}$, providing a Doppler shift from the absolute (radian) frequency

$$\omega = \sigma + \boldsymbol{k}.\boldsymbol{U} \tag{3}$$

While WW3 internally computes the wavenumber spectrum $F(\boldsymbol{k},\boldsymbol{x},t)$ due to its invariance properties, for output purposes this is converted to the traditional frequency-direction spectrum

$$F(f,\theta,\mathbf{x},t) = \left(\frac{2\pi}{c_g}\right) F(\mathbf{k},\mathbf{x},t) \tag{4}$$

The terms on the left-hand side of (1) represent spatial advection and the shifts in wavenumber magnitude and direction due to refraction by currents and varying water depth. The source term $S$ on the right-hand side of (1) represents all other processes that transfer energy to and from wave spectral components. It can be expressed as

$$S = S_{in} + S_{ds} + S_{nl} + S_{ice} + ... \tag{5}$$

to include contributions from wind forcing ($S_{in}$), energy dissipation ($S_{ds}$), weakly nonlinear four-wave interactions ($S_{nl}$), scattering and wave-ice interactions ($S_{ice}$), and other terms.





### 2.2.1 Wave forecast implementations

The wave component of EcoConnect consists of three nested implementations of WW3:

1. The global model (GLOBALWAVE) implemented on a regular latitude-longitude grid covering longitudes 0° to 360° at 0.234375° (<26 km) resolution, and latitudes -81.25° to +81.25° at 0.15625° (∼17 km) resolution. Atmospheric inputs are provided by the UKMO (Maher and Earnshaw, 2022). GLOBALWAVE forecast operates twice-daily (00 UTC and 12 UTC) to provide 6-day forecasts. Daily ice concentration fields are also sourced from the UKMO. No current or sea level inputs are used. A global time step of 900 s is used, with minimum time steps of 180 s for spatial advection and source term integration, and 900 s for refraction.

2. The regional model (NZWAVE) is implemented on a regular latitude-longitude grid covering longitudes 143.3203125° to 184.5703125° at 0.05859375° (<6 km) resolution, and latitudes -54.45° to -20.85° at 0.0390625° (∼4 km) resolution (refer to Fig. 1). Atmospheric inputs are provided by NIWA's deterministic forecast implementation of the Unified Model for the Tasman Sea and New Zealand (NZLAM, see section 2.3 for details), with both models running four times daily (00, 06, 12 and 18 UT) 76-hour forecasts. No ice, current or sea level inputs are used. Spectral boundary conditions for
NZWAVE are sourced by GLOBALWAVE. A global time step of 300 s is used, with minimum time steps of 60 s for spatial advection and source term integration, and 300 s for refraction.

3. The mainland Aotearoa New Zealand model (NZWAVE-HR) is implemented on a regular latitude-longitude grid covering longitudes 163.21° to 181.67° at 0.029296875° (<3.25 km) resolution, and latitudes -48.54° to -30.84° at 0.019531250° (∼2 km) resolution (Fig. 1). Atmospheric inputs are provided by NIWA's deterministic convection-resolving forecast im-
plementation of the Unified Model (NZCSM, see section 2.3 for details), with both models running four times daily (00, 06, 12 and 18 UT) 48-hour forecasts. Storm surge forecasts of sea level and current fields (NZSURGE-HR, forced by NZCSM) are combined with tidal sea levels and currents derived from the NZTIDE harmonic tidal model to provide input fields for NZWAVE-HR (more details about NZSURGE-HR and NZTIDE in section 2.4). Spectral lateral conditions for NZWAVE-HR are provided by NZWAVE. A global time step of 180 s is used, with minimum time steps of 60 s for
spatial advection, 30 s for source term integration, and 180 s for refraction.

### 2.3 Atmospheric Forcing

Atmospheric forcing for the wave models are derived from UKMO (global model, Maher and Earnshaw, 2022) and NIWA's family of numerical weather prediction (NWP) models in EcoConnect. UKMO provides 10-m winds and sea ice concentration for GLOBALWAVE. Whereas, NZWAVE and NZWAVE-HR are forced by near surface winds from the NIWA's regional
models. NIWA operates two limited area NWP models: i) the NZLAM, which featured a 12 km horizontal resolution between 2007 and late 2019, and since then at 4.4 km horizontal resolution, and ii) the NZCSM, which runs with a 1.5 km horizontal resolution. NZLAM's domain is represented by the larger black outline in Fig. 1, and NZCSM's domain is the smaller outline in Fig. 1, covering Aotearoa New Zealand's main landmass and its coastal waters.





Both models are based on the UM, a non-hydrostatic, fully compressible, deep-atmosphere model whose dynamical core,
ENDGame (Even Newer Dynamics for General atmospheric modelling of the environment), solves the equations of motion
using mass-conservation, semi-implicit, semi-Lagrangian, time-integration methods (Wood et al., 2014). However, they feature
differing science configurations and workflow set ups. NZLAM's workflow includes a 3-dimensional variational (3D-Var) data
assimilation method which ingests observations from satellites, aircraft, ships, buoys and land surface synoptic weather stations,
and is configured to use the Met Office GA6 Global model science settings (Walters et al., 2017). NZLAM forces NZWAVE
with outputs of 1-hour temporal resolution and the forecast extends 72 hours into the future. It runs four times a day at 00, 06,
12 and 18 UTC generating analyses at each cycle.

NZCSM is a convection-permitting model with minor changes to NZLAM configuration that improve model stability to run
over Aotearoa New Zealand's complex terrain with high resolution in a smaller domain. The scientific configuration of NZCSM
is equivalent to the set up described in Bush et al. (2020). Like NZLAM, NZCSM is warm-cycled, it restarts from an output
155  of the previous forecast. However, NZCSM does not perform its own data assimilation. Instead, at the start of each forecast
cycle, the larger-scale analysis from NZLAM, which has benefited from its data assimilation, is merged with the forecast from
the previous NZCSM cycle to give an improved atmospheric state from which another forecast can begin. NZCSM's lateral
boundary conditions are derived from NZLAM with a 20 minute update interval and operates 4 times daily on the 00, 06, 12 and
18 UTC analysis cycle, forecasting 48 h ahead. Forecast outputs from NZCSM, including the driving data for the downstream
wave models and other components in EcoConnect, are made available at 30 minute temporal resolution.

## 2.4  Water level and velocity forcing

Tidal and storm surge (infra-inertial) variability of water level and depth-averaged velocity are predicted using the RiCOM
(River and Coastal Ocean Model), a semi-implicit semi-Lagrangian finite element model based on an unstructured triangu-
lar grid that can be run both as a harmonic solver and a time-stepping hydrodynamic solver (Walters, 1992, 2005, 2006).
Tides (NZTIDE) and storm surge (NZSURGE and NZSURGE-HR) are calculated separately within EcoConnect for increased
computation efficiency. Simulating barotropic tides would require smaller time steps; instead, they are resolved harmonically,
saving computational time. These physical components are then summed to give total water level and velocity.

The hydrodynamic part of the RiCOM solves the Reynolds-Averaged Navier Stokes Equations assuming the incompressibil-
ity condition. A semi-implicit time-stepping scheme is used to solve advection using a semi-Lagrangian algorithm (Staniforth
and Côté, 1991) including a power series particle tracking method (Walters et al., 2007). The Coriolis term is added explicitly
using a third-order Admas-Bashforth scheme (Walters et al., 2009).

Two storm surge forecasts (NZSURGE and NZSURGE-HR) are included in EcoConnect, both running 4 times daily using
the time-stepping RiCOM model forced by 10-m winds and mean sea level pressure but sourcing these inputs from the 72-hour
NZLAM and 48-hour NZCSM forecasts, respectively. The new atmospheric forcing file is smoothed from the previous file over
the first 6 hours of the forecast to account for changes in the atmospheric variables due to data assimilation. Wind forcing is
included in the model as surface stress using a quadratic formulation based on wind velocity with a drag coefficient specified by
Wu (1982). Surface pressure is used to calculate an inverse barometer surface level which is applied as a loading term. Relaxing





to an inverse barometer water level within a radiation boundary condition is applied as the lateral open boundary conditions. Further details of the NZSURGE operational model can be found in Lane et al. (2009) and verification of the results in Lane and Walters (2009).

The tidal model (NZTIDE) is calculated using a harmonic tidal model formulation of RiCOM (Walters et al., 2001) based on a harmonic decomposition in time and a finite element approximation in space. This model provides amplitudes and phases for the eight largest tidal constituents around Aotearoa New Zealand: M2, N2, S2, K2, K1, O1, P1 and Q1. Dependent variables are expressed in terms of harmonic expansions of these constituents with the non-linear bottom friction included as a series expansion (Walters, 1992). Equilibrium tide and self-attraction/loading tide are also included in the formulation (Goring and Walters, 2002). The results of the tide model are interpolated from the unstructured grid onto a regular Cartesian grid for simplified viewing and outputs. For each forecast period, the tidal sea surface and currents are reconstructed from these tidal constituents. Tides in Aotearoa New Zealand are dominated by the M2 constituent which proceeds around the two main islands in an anticlockwise direction. Tidal flows through Cook Strait (the narrow Strait between Te Ika A Maui/North Island and Te Waipounamu/South Island) can be especially strong because the tide levels are close to 180° out of phase at either end. NZTIDE and NZSURGE-HR water levels and currents are summed and interpolated onto the NZWAVE-HR domain to provide total water level and velocity forcing for that wave model.

A new tidal and storm surge model has been developed using TELEMAC, a finite element model simulates tides and storm surge simultaneously using a time-stepping approach (for technical details about the model please refer to Hervouet, 2000; Moulinec et al., 2011). This new model is forced by UKMO global model. We extracted water level and velocity output from this model at Baring Head station for wave-current interaction analysis. This new model is planned to replace RiCOM as the operational water level and velocity forecasting model.

### 2.5 Observations

#### 2.5.1 Satellite data

Near-real-time, gridded, satellite derived daily average significant wave heights from CMEMS were used to validate the forecasts between 1st of Jan to 31st of Dec 2021. This gridded product includes daily mean and maximum significant wave height data from different altimeter missions. This product merges along-track data from Jason-3, Sentinel-3A, Sentinel-3B, SARAL/AltiKa, Cryosat-2, CFOSAT, and HaiYang-2B missions and delivers daily data at 2° spatial resolution with an uncertainty ranging from 0.12 to 0.44 m (CMEMS, 2024, last access on the 11th of April 2024).

#### 2.5.2 Buoy data

Two *in situ* sites with near real-time wave observations are used to validate the predicted wave height, mean period and peak direction. The Banks Peninsula wave buoy is a directional wave rider moored approximately 17 km east of Steep Head, Banks Peninsula in Te Waiponamou/South Island at latitude 43°45' south, longitude 173° 20' east, in approximately 80 m of water depth (refer to Fig. 2a). The mooring location has been continuously maintained since 1999 with data gaps limited to





buoy/mooring failures. The location is exposed to a wide range of swells from the northeast to the south. The mean significant wave height over the observation period is 2.1 m with an average mean period (Tm02) of 6.5 s (Bosserelle, 2022).

The Baring Head station has been instrumented since 1995 with non-directional wave rider buoy at the beginning and with directional wave rider buoys since 2014. The site is located in approximately 44 m depth, 2 km west of Baring Head lighthouse and 15 km south-east of Te Whanganui-a-Tara/Wellington. The site is located inside the Cook Strait and is sheltered from many wave directions by both Te Waiponamou/South Island and Te Ika A Maui/North island. Most swells come from the South but the location is also exposed to a narrow swell window from the west/northwest (Allis et al., 2021).

Observations from the 1st of Jan 2021 to 31st of Dec 2021 from the two stations are used in this study. Before this period both stations switched from the second (Tm02) to the first order mean period (Tm01). Time series of significant wave height, first order mean period (Tm01), and peak direction are used for model validation. Wave observations are available every 15 minutes and are filtered using a 1-hour moving average filter. Significant wave height and peak direction are decomposed into u and v components, filtered (1-hour moving mean), then recomposed to remove spiky variability in both time series. The locations of wave buoy measurements are shown in Fig 2, as well as NZWAVE-HR bathymetry and GLOBALWAVE and NZWAVE 30- and 80-m isobaths. Here one can expect that the bathymetry resolution will generate differences in how wave refraction will occur in the nearshore, especially for long-period swells.

## 2.6 Experiment design and evaluation metrics

Five numerical wave forecasts are evaluated in this study. Three simulations are part of EcoConnect's forecast: GLOBAL-WAVE, NZWAVE and NZWAVE-HR. The latter one includes tidal and storm surge forcing via water level and currents inputs to the model – hereinafter tides. All models use a calibrated version of ST4 physics (Gorman and Oliver, 2018). The operational forecasts have run operationally in the current cluster since 2018 and are analysed in this study.

Two additional forecasts are run to address the questions raised in this study. In one simulation we use ST6 physics with default parameters (Table 2.8 in WW3DG, 2019) applied to the intermediate domain and the forecast experiment is called NZWAVE-ST6. This experiment sheds light on ST6 performance around Aotearoa New Zealand waters for later consideration of these source terms in EcoConnect's forecasts. The last experiment removed tides and storm surge forcing from the highest resolution wave model and is called NZWAVE-HR-NOTIDES. Comparisons between NZWAVE-HR-NOTIDES and NZWAVE-HR shows the importance of ocean currents as forcing to generate wave variability in coastal Aotearoa New Zeland. The two additional forecasts start on the 1st of Dec 2020 from rest and they have a spin-up period of one month – similar to Gorman and Oliver (2018).

We use 24-hour forecasts starting at 00:00 UTC daily. Model results are interpolated to the satellite observational field and the closest model grid point is used for comparison against *in situ* data. Forecast outputs are written half-hourly for the highest resolution models (NZWAVE-HR and NZWAVE-HR-NOTIDES), and hourly for the rest of the models. Model outputs are linearly interpolated to the observational frequency before model-data comparison. Model significant wave height daily averages are validated using satellite data. Model time series of significant wave height (Hs), peak period (Dp), and first-



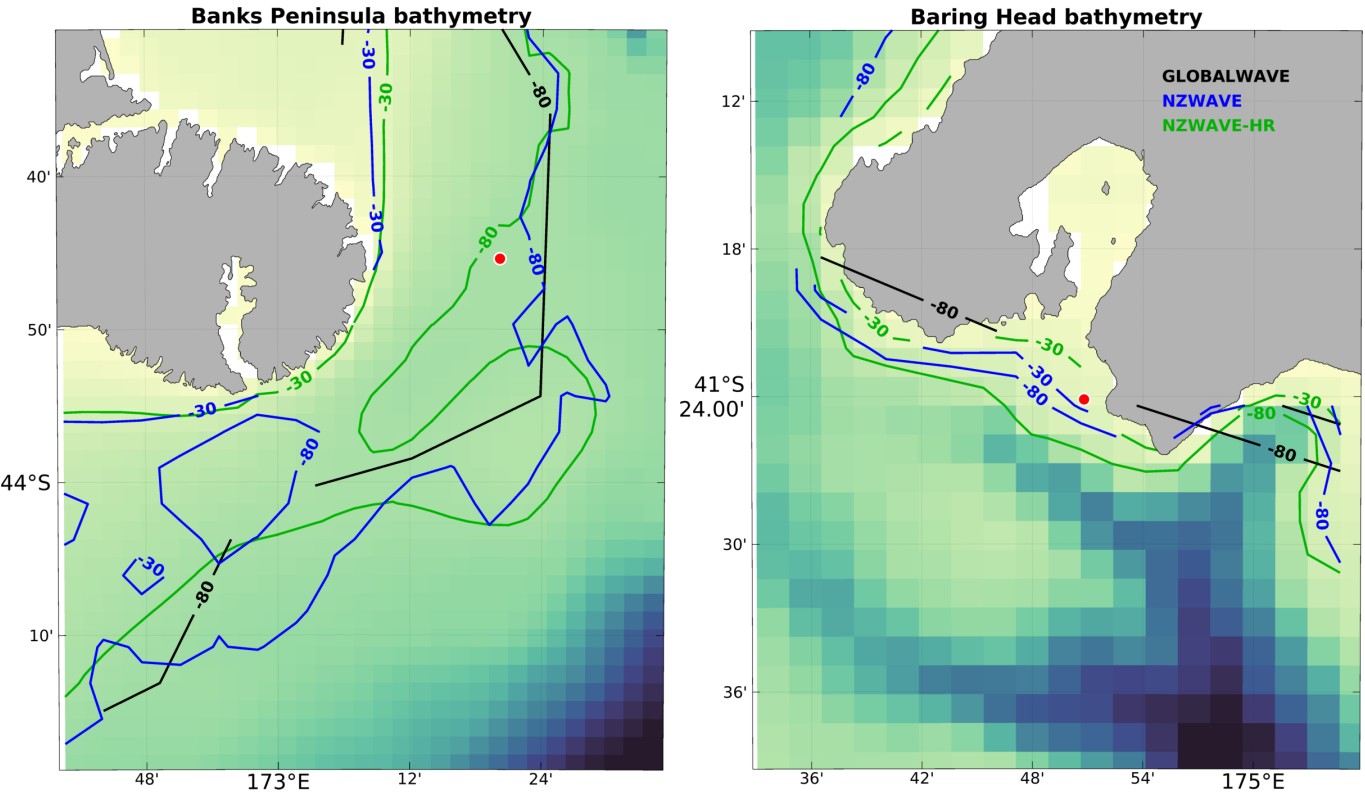

**Figure 2.** Model bathymetries at Banks Peninsula (a) and Baring Head (b) showing locations of wave buoy measurements (red dots). NZWAVE-HR bathymetry is displayed using the colour shade and green contour. The 30- and 80-m isobaths from NZWAVE (blue) and GLOBALWAVE (black) are also shown for comparison.

order mean period (Tm01) are also validated at Banks Peninsula and Baring Head coastal stations (Fig. 2). The forecasts are objectively validated using the root mean square error (RMSE) given by:

$$\text{RMSE} = \sqrt{\frac{1}{n}\Sigma_{i=1}^{n}(x_i - y_i)^2}; \tag{6}$$

and linear correlation (r):

$$\text{r} = \frac{\sum_{i=1}^{n}(x_i - \bar{x})(y_i - \bar{y})}{\sqrt{\sum_{i=1}^{n}(x_i - \bar{x})^2}\sqrt{\sum_{i=1}^{n}(y_i - \bar{y})^2}}; \tag{7}$$

between observed ($x$) and predicted ($y$) results, where $i=1,2,...,n$ are the observation times or locations and the averages ¯ are applied in time or space. Spatial fields of significant wave height RMSE and bias (model - observations) are computed using

daily averaged model results interpolated to satellite observations horizontal grid.





Harmonic analysis is applied to *in situ* observations and NZWAVE-HR forecast time series of significant wave height, mean period and peak direction using t_tide (Pawlowicz et al., 2002). This allows us to identify tidal oscillations in these wave parameters.

## 3   Results

This section analyses the impacts of downscaling, source terms, and tides on forecasting significant wave height, wave peak direction and mean period (Tm01). Initially, the historical 24-hour forecasts are evaluated against satellite data, and *in situ* observations from Banks Peninsula (open coast) and Baring Head (constricted region) using RMSE, linear correlation analysis and bias metrics. This section is concluded by highlighting the impacts of the tides and storm surge on the wave parameters, especially at Baring Head.

### 3.1   Mean spatial fields

Mean bias and RMSE analyses between altimeter data and the historical wave forecasts are given in Fig. 3 and 4. The 1-year average of satellite observations indicates an area with large significant wave height ($\sim$4.5 m ) below 48°S which penetrates the Tasman Sea and offshore regions of southeast Aotearoa New Zealand (Fig. 3a). The northeast region of Te Ika a Maui/North Island is marked with the smallest average waves due to the landmass barrier that protects the region from south/southwest

swells. All forecasts using the calibrated source term 4 (ST4) (Gorman and Oliver, 2018) show a relatively larger positive mean bias of significant wave height on the west side of Aotearoa New Zealand, whereas the bias is small on the eastern side of the country (Fig. 3b,c,e,f). The spatial mean bias for the smallest domain (NZWAVE-HR) varies from 0.10 to 0.13 m between all ST4 forecasts. Applying ST6 physics (NZWAVE-ST6) with default parameters reduces the positive bias on the western side of Aotearoa New Zealand compared to the same model using calibrated ST4 (NZWAVE) (Fig. 3c,d). However, it also produces

a negative bias on the eastern side of the country, generating a mean bias of -0.07 m computed for the smallest domain. Removing tides from the forecast (NZWAVE-HR-NOTIDES) shows no significant change in the mean bias field analysis (Fig. 3e,f). Implying that tides and storm surge forcing do not significantly affect the temporal mean values of these wave parameters.

In Fig. 4 1-year significant wave height standard deviation from altimeter data is shown alongside the RMSE comparisons with the various historical forecasts described in Section 2.2. There is considerable variability in wave height south of Tas-

mania and the Re Waipounamu/South Island of Aotearoa New Zealand in which large standard deviations of significant wave height ($\sim$1.5 m) are observed in these regions (Fig. 4a). This region of large variability fades towards the north and reaches minima values ($\sim$0.75 m) above 30°S and on the eastern side of Aotearoa New Zealand. GLOBALWAVE and NZWAVE show similar RMSE in the whole analysed region and for the smallest domain (Fig. 4b,c). This is associated with the same ST4 parameters used in both models. Model resolution does not significantly impact these results and GLOBALWAVE, NZWAVE

and NZWAVE-HR-NOTIDES have similar mean RMSE. However, satellite observations have low spatial resolution (2°) and some coastal points show larger RMSE which are associated with localised larger uncertainty in the satellite observations.





**Figure 3.** Maps of mean observed significant wave height (m) and forecast bias (m) from GLOBALWAVE (b), NZWAVE (c), NZWAVE-ST6 (d), NZWAVE-HR-NOTIDES (e) and NZWAVE-HR (f) models. Statistics are computed between between 1st of Jan to 31st of Dec 2021. The spatial means are shown in the figure title and highlighted over defined regions.





Applying ST6 default parameters (NZWAVE-ST6) reduces model RMSE south of Tasmania and minimises the mean RMSE from 0.43 to 0.41 m compared to NZWAVE for their whole domain. NZWAVE-ST6 also decreases significant wave height RMSE northwest of the North Island but increases it on the east side of the South Island. In general, ST6 physics reduces the mean RMSE in the NZWAVE-HR region by 0.01 m (Fig. 4c,d). Including tides as forcing in the wave model (NZWAVE-HR) slightly increased (0.01 m) the average RMSE of significant wave height.

### 3.2 *In situ* significant Wave Height (Hs)

Model evaluation against *in situ* significant wave height (Hs) shows similar RMSE (0.30–0.35 m) and correlation coefficient (0.91–0.92) between the sensitivity experiments at Banks Peninsula – an open coast station (Fig. 5). However, relatively large differences are found when analysing Hs bias. A reduction of the negative bias from -0.18 m (GLOBALWAVE) to -0.08 m (NZWAVE), and an even further reduction (-0.04 m) is found in NZWAVE-HR-NOTIDES (Fig. 5a,b,d). Including tides in the simulation (NZWAVE-HR) does not generate a marked impact on evaluation metrics (RMSE, bias and r) and shows a slight degradation in bias (Fig. 5d,e). Using standard ST6 parameters (NZWAVE-ST6) degraded the forecast bias (-0.15 m) and RMSE (0.35 m) compared to the calibrated ST4 forecast (NZWAVE) (Fig. 5b,c).

Significant wave heights smaller than 2 m have larger probability density estimates in the forecasts and observations. Wave heights between 2 and 6 m tend to have an even distribution in regards to over- and underestimation of the observed values. Some events with wave height between 4 and 6 m are overestimated by the forecasts and fall within the 6 to 8-m category. Within the observed 6 to 8-m category, most forecasted waves are within the same range except for a few data points that fall into the 4 to 6-m category. The largest waves (∼8 m) were underestimated by GLOBALWAVE and NZWAVE-ST6.

At Baring Head, a constricted coastal region, larger improvements in the evaluation metrics are generated by the downscaling in comparison to Banks Peninsula (Fig. 6). A decrease of 25% (0.09 m) in significant wave height RMSE is shown from the lowest resolution (GLOBALWAVE) to the highest resolution model (NZWAVE-HR-NOTIDES) (Fig. 6a,d). The intermediate grid generates a 17% (0.06 m) decrease in RMSE compared to GLOBALWAVE (Fig. 6a,b). However, applying standard ST6 parameters to the intermediate model domain (NZWAVE-ST6) reduced the forecast skill (RMSE and r) back to the level of the GLOBALWAVE forecast (Fig. 6a,c). Including tides and storm surge forcing into the highest resolution model (NZWAVE-HR) generates a slight improvement in forecast RMSE and correlation coefficient (Fig. 6d,e).

Significant wave height forecast bias switches from positive (GLOBALWAVE = 0.13 m) to negative (NZWAVE = -0.12 m) in the first downscale exercise but it is improved (-0.05 m) in the highest resolution model (NZWAVE-HR-NOTIDES). The largest positive bias in GLOBALWAVE might be related to the reduced levels of wave height dissipation in the region due to the poor representation of the coastline and bathymetry near Baring Head (Fig. 2b). The negative wave height bias in NZWAVE (-0.12 m) is further reduced in NZWAVE-HR-NOTIDES (-0.05 m) which better represents the underwater topography and coastline.

Wave heights tend to be smaller (∼1 m) at Baring Head compared to Banks Peninsula (∼2 m). Its probability density estimate is the largest near 1-m wave height in all forecasts. The region with highest probability density is located around the





**Figure 4.** Maps of observed significant wave height standard deviation (a) in metres and forecast RMSE (m) from GLOBALWAVE (b), NZWAVE (c), NZWAVE-ST6 (d), NZWAVE-HR-NOTIDES (e) and NZWAVE-HR (f) models. Statistics are computed between the 1st of Jan to 31st of Dec 2021. The spatial means are shown in the figure title and highlighted over defined regions in metres.



**Banks Peninsula**

**Figure 5.** Scatter plots of observed and forecasted significant wave heights from GLOBALWAVE (a), NZWAVE (b), NZWAVE-ST6 (c), NZWAVE-HR-NOTIDES (d) and NZWAVE-HR (e) simulations at Banks Peninsula station. Root mean square error (RMSE), mean bias and coefficient of correlation (r) are shown at the top-left corner of each panel. The scatter colour represents the probability density estimate.

315  1:1 line for most forecasts except GLOBALWAVE. The latter forecast has a larger density estimate above the 1:1 line which represents overestimation for most of those small wave events.





**Figure 6.** Scatter plots of observed and forecasted significant wave heights from GLOBALWAVE (a), NZWAVE (b), NZWAVE-ST6 (c), NZWAVE-HR-NOTIDES (d) and NZWAVE-HR (e) runs at Baring Head station. Root mean square error (RMSE), mean bias and correlation coefficient (r) are shown at the top-left corner of each panel. The scatter colour represents the probability density estimate.





### 3.3 *In situ* mean period (Tm01)

Model evaluation analysis of the first-order mean period (Tm01) shows root mean square error (RMSE) and absolute bias smaller than 0.8 s and 0.3 s, respectively, for all forecasts (Fig. 7). All models had similar and high correlation coefficients with observations of Tm01 (>0.8). The most marked changes occur between NZWAVE-ST6 and NZWAVE. A small degradation of the forecast RMSE and correlation coefficient is found when comparing NZWAVE-ST6 (RMSE=0.77 s, r=0.85) and NZWAVE (RMSE=0.65 s, r=0.89). However, a smaller absolute bias is simulated by the former model (-0.04 s) compared to the latter forecast (-0.14 s). A Tm01 gradual bias reduction is found from the lowest resolution (GLOBALWAVE, bias=-0.28 s) to the highest resolution forecast without tides (NZWAVE-HR-NOTIDES, bias=-0.10 s). Including tides and storm surge forcing slightly degraded the Tm01 bias (NZWAVE-HR, bias=-0.11 s) compared to the same model without varying water level and current forcing (NZWAVE-HR-NOTIDES, bias=-0.10 s).

Probability density estimate analysis of first-order mean wave period (Tm01) shows a distribution of predicted and observed values generally dispersed between 4 and 14 s for all models (Fig. 7). A relatively larger density estimate (∼0.10) occurs for Tm01 between 5 and 9 s, which is situated around the 1:1 ratio showing that most predicted Tm01 is similar to observed values for all forecasts.

Model evaluation statistics of mean period show larger error in Baring Head compared to Banks Peninsula. Maximum Tm01 forecast RMSE (NZWAVE-ST6 = 1.75 s) almost doubled when comparing the two stations. GLOBALWAVE has the smallest Tm01 forecast RMSE (1.46 s) and largest correlation coefficient (r=0.85). The downscaled models (NZWAVE and NZWAVE-HR-NOTIDES) increased RMSE and reduced the correlation coefficient compared to GLOBALWAVE. However, a gradual improvement in Tm01 bias from GLOBALWAVE (-0.78 s) to NZWAVE (-0.67 s), and NZWAVE-HR-NOTIDES (-0.35 s) is seen. Moreover, adding tidal and storm surge forcing in NZWAVE-HR slightly reduced RMSE (1.63 s) and absolute bias (-0.34 s). The probability density estimate shows larger values of Tm01 concentrated between 4 and 11 s for all forecasts. However, this region of higher density estimate lies below the 1:1 curve for all models which points out the negative biases in all forecasts (Fig. 8).

### 3.4 *In situ* peak wave direction

At Banks Peninsula station, the observed peak wave direction is largely dominated by the south component (∼30% of occurrence) with significant wave height reaching more than 6 m (Fig. 9a). Large waves (>6 m) also arrive from the SSW quadrant, however, this direction has a smaller percentage of occurrence (∼5%). The second most frequent direction is the SSE (∼15%) and is followed by an eastern component (∼13%). This eastern component, however, is dominated by smaller waves.

The numerical simulations show a wider spread in larger waves arriving from the S and SSW quadrants at Banks Peninsula station (Fig. 9). That differs from the observations which have the largest frequency of occurrence more concentrated on the S component. GLOBALWAVE and NZWAVE-ST6 have an even distribution between those two directions, whereas NZWAVE shows a slightly larger preferential distribution towards the S component. The highest-resolution models, however, show larger occurrences of the SSW direction component. All simulations show a marked northeast component with the percentage of



**Figure 7.** Scatter plots of observed and forecasted first order mean period (Tm01) from GLOBALWAVE (a), NZWAVE (b), NZWAVE-ST6 (c), NZWAVE-HR-NOTIDES (d) and NZWAVE-HR (e) runs at Banks Peninsula station. Root mean square error (RMSE), mean bias and coefficient of correlation (r) are shown at the top-left corner of each panel. The scatter colour represents the probability density estimate. Note the shrunken colour axis.

occurrence ranging from ~10% (GLOBALWAVE) to ~15% (NZWAVE) with wave height reaching values above 4 m. This



**Figure 8.** Scatter plots of observed and forecasted first order mean period (Tm01) from GLOBALWAVE (a), NZWAVE (b), NZWAVE-ST6 (c), NZWAVE-HR-NOTIDES (d) and NZWAVE-HR (e) runs at Banks Peninsula station. Root mean square error (RMSE), mean bias and coefficient of correlation (r) are shown at the bottom-right corner of each panel. The scatter colour represents the probability density estimate. Note the shrunken colour axis.





large-wave (>4 m) northeastern component is also observed in the buoy measurements, however, with a smaller percentage of occurrence (∼12%).

**Figure 9.** Directional histograms (wave roses) of peak wave direction and significant wave height from buoy measurements (a), GLOBAL-WAVE (b), NZWAVE (c), NZWAVE-ST6 (d), NZWAVE-HR-NOTIDES (e) and NZWAVE-HR (f) runs at Banks Peninsula station.





Peak wave direction observations have the largest frequency of occurrence from the SSE direction at Baring Head station
(Fig. 10a). This might be associated with strong tidal currents that occur in the region with a marked southeastward residual
component (Fig. 7 of Walters et al., 2010) which steer incoming southerly swells towards a similar current direction (more
details of the process are discussed in section 3.5). Waves from the south quadrant are the second most frequent. Both SSE and
S direction can have the largest significant wave heights (>6 m). In contrast, the NW component (3$^{\text{rd}}$ most frequent) is marked
by smaller waves (<1 m) locally generated by the strong NW winds that often happen in the region (Reid, 1996).

All forecasting models show a prevalence of waves coming from the south (Fig. 10). The percentage of occurrence ranges
from 52% (GLOBALWAVE) to 58% (NZWAVE-HR). The second most common peak wave direction is the SSW. Both S and
SSW components can have waves larger than 4 m, but waves between 1 and 2 m are the most frequent. NZWAVE is the only
model that has its third largest component associated with the NW direction – similar to observations. GLOBALWAVE and
NZWAVE-ST6 have NNW as their third most frequent direction. The highest-resolution models (NZWAVE-HR and NZWAVE-
HR-NOTIDES), however, show a SE component to be the third most frequent. The lack of a NW/NNW component in those
models might be associated with weaker north-westerlies in the atmospheric forcing (NZCSM) which are not able to generate
the observed smaller (<1 m) and frequent waves in the region.

### 3.5 Tidal influence on wave height, period and direction

A close look at intra-daily variability shows high-frequency oscillations in the observed significant wave height and mean pe-
riod (Tm01) which are often matched by ∼12-hourly peaks in predicted wave height and period by NZWAVE-HR (Fig. 11a,b).
This 12-h variability simulated by NZWAVE-HR seems to be generated by the interaction between waves and tidal currents
(Fig. 11). This process has been observed and simulated in different regions of the globe (e.g., Ardhuin et al., 2012; Rapizo
et al., 2017; Barnes and Rautenbach, 2020; Halsne et al., 2024). Wave buoy measurements have additional high-frequency os-
cillations (green dots in Fig. 11a,b) not accounted for by NZWAVE-HR due to its limitations in model approximations and/or
in its forcings. Similar behaviour has been found in wave variability that has been linked to the resolution of atmospheric
forcing in embayments with complex orographic features (Daniels et al., 2022). Salonen and Rautenbach (2021) also indicated
that bathymetric features, like underwater mounds, could significantly modify the nearshore wave climate if these features
were not adequately resolved in the model bathymetry. In this study, the forecast model without tidal and storm surge forc-
ing (NZWAVE-HR-NOTIDES) does not show any marked intra-daily variability in significant wave height and mean period
(Tm01) which highlights the importance of including tides as forcing (Fig. **??**).
At Banks Peninsula, NZWAVE-HR simulates this 12-h variability that occurs due to the interaction between southerly
waves and the anti-clockwise propagation of the M2 tide on the continental shelf. Around midnight on the 21st and 22nd of
Oct 2021, southward (negative) tidal currents flow against a swell propagating northeastward (20°) (red rectangles in Fig. 11).
These counter-currents increase wave height and reduce the wave period every tidal cycle near low tide which propagates on
the continental shelf as a progressive wave – peaks in water level match peaks in tidal currents (Walters et al., 2001). This
growth in significant wave height and shortening in wave period while facing opposing currents is well predicted in theory,
observed and simulated in different regions (Phillips, 1977; Vincent, 1979; Ardhuin et al., 2012; Rapizo et al., 2017; Barnes







**Figure 10.** Directional histograms (wave roses) of significant wave height from buoys measurements (a), GLOBALWAVE (b), NZWAVE (c), NZWAVE-ST6 (d), NZWAVE-HR-NOTIDES (e) and NZWAVE-HR (f) runs at Baring Head station.

and Rautenbach, 2020; Halsne et al., 2024). The reduction in wave period/length is called "wave straining" by Holthuijsen and Tolman (1991). It is a combination of the "concertina effect" (Ardhuin et al., 2017; Wang and Sheng, 2018), related to changes



in wavelength, and the "energy bunching" (Baschek, 2005). Wave peak direction has values near 200° and tidal variability is
virtually absent between the 20th and 23rd of Oct 2021 (not shown).

**Figure 11.** Timeseries of significant wave height (a), first-order mean period (Tm01) and depth-averaged meridional currents (c) at Banks
Peninsula from buoy measurements (green), NZWAVE-HR-NOTIDES (cyan) and NZWAVE-HR (blue).

Hourly maps of the ocean current field reveal tidally-driven variability in significant wave height at Banks Peninsula station
(Fig. 12). Large significant wave height from buoy observations (2.09 m) and the NZWAVE-HR forecast (2.19 m) coincide
with opposing southward currents at noon on the 21st of Oct 2021 (Fig. 12a). The swell decays throughout the rest of the day
but also oscillates with the currents as shown in Fig. 11. Observed and predicted significant wave height reach local minima of
1.90 m and 1.89 m at 21h on the 21st of Oct 2021, when currents are the weakest (Fig. 12j). Measured and simulated significant





wave height increase again with opposing currents at 22 h on the 21st of Oct 2024 and increase to local maxima at 23 h of the same day.

Tidal forcing plays a larger role at Baring Head due to its location in a region of constricted circulation with strong tidal currents through Cook Strait. Large variability is observed in peak wave direction and significant wave height (Fig. 13). Around
midnight and noon on the 13th of January 2021, southeastward (counter) currents (red rectangles in Fig. 13) increase significant wave height (by around 10 cm) and steer their direction to arrive from 140°. The opposite happens when the currents flow northwestward and the waves decrease their height and shift direction to 220°, generating oscillations in tidal steering direction with an amplitude of 40°. NZWAVE-HR simulates this tidal variability but with smaller amplitudes of significant wave height (5 cm) and peak wave direction (6°). NZWAVE-HR-NOTIDES, however, does not show any 12-h variability in significant
wave height or peak direction. Despite the smaller tidal amplitudes in wave height and direction compared to observations, they explain the slight improvements in forecasted significant wave height and mean period (Tm01) at Baring Head (Fig. 6 and 8)

Tidal variability is also observed in the mean period (Tm01) which has a tidal amplitude of about 1.5 s, whereas the model has an amplitude of 0.5 s (not shown). The smaller tidal variability in the model can be explained by the smaller current speeds
in RiCOM (NZTIDE+NZSURGE-HR). TELEMAC, the newly developed storm tide forecast shows twice the current velocity generated by RiCOM at this location which would create larger tidal variability in NZWAVE-HR. For instance, analytical analysis conducted by Barnes and Rautenbach (2020) based on a simple model for wave refraction from zero velocity water into a steady current derived by Johnson (1947) shows that waves with a period of 12 s being acted on by a current field of 0.8 m/s at a 45° angle could shift another 30° due to wave refraction. This is a similar case to the observed waves at Baring Head.
Hourly maps of the velocity field show a large oscillation (up to 80°) in peak direction that occurs during the tidal cycle at Baring Head. At 3 am 13th of Jan 2021, the observed wave peak period has its smallest value (142.8°) when local currents are at their peak (Fig. 14c). NZWAVE-HR shows small levels of steering and peak direction is 188.4°. When currents are flowing northwestward at 7 am 13th of Jan 2021, observed peak period shows its daily maximum steering and tends to be orthogonal to the flow (Fig. 13 and 14g). The predicted wave peak direction switches about 5° and waves arrive from an angle of 193°. This
smaller amplitude in the tidally forced variability in peak direction is attributed to the hydrodynamic model's low resolution and/or harmonic solver which is not able to capture spatial and temporal variability in current speed.

Harmonic analysis of wave parameters reveals a marked influence of the main semi-diurnal tidal constituent (M2) on observed and forecast significant wave height and peak direction (Table 1). The same analysis is applied to mean period (Tm01) but the amplitudes are smaller than 1 s and are not shown. At Banks Peninsula, the observed significant wave height (Hs) and
peak direction show an M2 amplitude of 2.1 cm and 2.9° which are well represented by NZWAVE-HR (2.0 cm and 3.0°). The analysed wave parameters show larger tidally-driven amplitudes at Baring Head compared to Banks Peninsula. This happens due to Baring Head's geography which generates larger tidal currents, hence a greater influence on the wave parameters. The observed M2-forced variation in significant wave height is about 5 cm and is closely simulated by NZWAVE-HR (3.8 cm). The observed peak wave direction shows a large (17.1°) M2-forced oscillation which is not reproduced by NZWAVE-HR. This







**Figure 12.** 2-hourly maps of ocean currents in m/s (green shade and black arrows) from NZTIDE and observed (magenta) and NZWAVE-HR-predicted (blue) significant wave height and peak direction at different times on the 21st of October 2021 at Banks Peninsula. The values of observed and predicted (NZWAVE-HR) significant wave height and peak direction for each time frame are written for reference. Model wave variables are extracted from the closest grid point to the observations.





**Figure 13.** Timeseries of significant wave height (a), peak wave period and depth-averaged meridional currents (c) at Baring Head from buoy measurements (green), NZWAVE-HR-NOTIDES (cyan) and NZWAVE-HR (blue).

might be related to NZWAVE-HR's relatively low resolution (2 km), weaker currents in the ocean forcing and/or lack of a 2-way wave-current coupled system.

## 4   Discussion and Conclusions

We used a set of numerical wave forecasts to evaluate the importance of downscaling, source terms and tides on the wave forecast around Aotearoa New Zealand. Focus was given to two sites on the southeast coast of the country where we have



**Figure 14.** Hourly maps of ocean currents in m/s (green shade and black arrows) from NZTIDE and observed (magenta), and NZWAVE-HR-predicted (blue) significant wave height and peak direction at different times on the 13th of January 2021 at Baring Head. Observed and predicted (NZWAVE-HR) significant wave height and peak direction for each time frame are written for reference. Model wave variables are extracted from the closest grid point to the observations.





**Table 1.** Amplitudes of M2 tidal constituent for significant wave height (Hs) and peak direction from observations / NZWAVE-HR.

| Harmonic constituent | Banks Peninsula | | Baring Head | |
|---|---|---|---|---|
| | Hs (cm) | Dp (°) | Hs (cm) | Dp (o) |
| M2 | 2.1 / 2.0 | 2.9 / 3.0 | 5.2 / 3.8 | 17.1 / 3.7 |

long-term time series from wave buoys. At these locations, a thorough validation of significant wave height, mean period and peak direction was conducted and the impact of tides on the forecast was analysed.

Wave model downscaling showed a marked impact at a coastal scale, especially at Baring Head – a constricted coastal region. A reduction of 25% in significant wave height forecast error was achieved by downscaling from the low-resolution global model (GLOBALWAVE, RMSE = 0.36 m) to the highest-resolution model (NZWAVE-HR-NOTIDES, RMSE = 0.27

m) at Baring Head. At Banks Peninsula, however, model downscaling didn't show large impact on the wave forecast. This might be related to the station's geography which is located in an open coast continental shelf region and its wave conditions can be well-simulated by low-resolution models. Model downscaling generates a reduction in the bias of the absolute mean period (Tm01) at the two stations. At Banks Peninsula, mean absolute bias was reduced from -0.28 s (GLOBALWAVE) to -0.10 s (NZWAVE-HR-NOTIDES). At Baring Head, a larger reduction in the bias was found when comparing GLOBALWAVE (-0.78

s) to NZWAVE-HR-NOTIDES (-0.35 s) mean period (Tm01). Nevertheless, model downscaling didn't show a marked impact on the mean period RMSE. No marked difference in wave peak period was found when comparing the different downscaled models.

The use of ST6 default parameters improved significant wave height forecast on the west coast of Aotearoa New Zealand but degraded it on its east coast. Satellite comparisons show that a forecast using ST6 default parameters reduced significant wave

height RMSE 0.02 m (0.01 m) in the Southwest Pacific (Aotearoa New Zealand) region in comparison to a model forecast using calibrated ST4 parameters. It also decreased the positive bias generated by the calibrated ST4 model from 0.13 m to -0.04 m in the Southwest Pacific region but created a larger region of negative bias on the eastern side of Aotearoa New Zealand. This explains the degraded significant wave height forecast at Banks Peninsula and Baring Head when comparing NZWAVE-ST6 to NZWAVE. Gorman and Oliver (2018) used altimeter wave data to find the set of ST4 parameters that minimises significant

wave height RMSE via an iterative process. The same process can be applied to ST6 parameters to further reduce satellite significant wave height RMSE to a point in which NZWAVE-ST6 might improve its results against in situ observations. In addition, one may try to minimise the RMSE of in situ and satellite significant wave height altogether since altimeter data has a larger observational uncertainty (0.12 - 0.44 m) compared to buoy measurements (<0.10 m).

Tides and storm surge forcing showed a marked impact on the wave variability at Baring Head. This is explained by the site

location which is in a large tidally constricted region – Cook Strait. A southerly swell interacting with opposing tidal currents coming from the north generates an increase in wave height by around 10 cm, a decrease in mean period of around 1.5 s and a shift in wave direction of around 40°. When the tidal current is flowing northward, the wave parameters are affected in the opposite sense. The high-resolution tidally-driven model (NZWAVE-HR) generates amplitudes of 5 cm (significant





wave height), 0.5 s (mean period) and 6° (peak direction). This simulation reduced the significant wave height RMSE from
0.27 to 0.26 m and increased the correlation coefficient from 0.94 (NZWAVE-HR-NOTIDES) to 0.95 (NZWAVE-HR). These
improvements are smaller in comparison to the decrease in error generated by the downscaling. This can be explained by the
long timeseries used in the study which mixes periods where tides are more and less important – spring-neap cycle. Another
factor is the weaker currents generated by the RiCOM tide and storm surge models which underestimate the impact of currents
on the wave field. At Banks Peninsula, a similar wave-current interaction process occurs but tides didn't show large impact
on the wave forecast evaluation statistics. This might be related to the station's geography which is located in an open coast
continental shelf region and its wave conditions were not largely affected by tidal currents.

The results found by our 2-km resolution tidally-driven wave forecast model (NZWAVE-HR) are comparable to 1-km reso-
lution simulation generated by Albuquerque et al. (2021). Those authors ran a wave hindcast for Aotearoa New Zealand using
SWAN and forced by the Climate Forecast System Reanalysis (CFSR, Saha et al., 2010, 2014). At Baring Head, they found
significant wave height hindcast bias, correlation coefficient and RMSE values to be -0.09 m, 0.87 and 0.16 m compared to
-0.05 m, 0.95 and 0.26 m (NZWAVE-HR). The increased correlation in NZWAVE-HR might be related to the high spatial and
temporal resolution atmospheric forcing, as well as the additional tidal input into the forecast. Nevertheless, the larger RMSE
might be attributed to the lower spatial resolution (2 km) in NZWAVE-HR compared to the 1-km grid spacing in the hindcast.
At Banks Peninsula, similar results were also found when comparing NZWAVE-HR to the wave hindcast generated by (Al-
buquerque et al., 2021). Hindcast significant wave height bias, correlation coefficient and RMSE were 0.05 m, 0.88 and 0.14
m compared to -0.06 m, 0.92 and 0.30 m (NZWAVE-HR). This comparison suggests that model calibration using *in situ* data
and/or increased model resolution are still needed to further reduce model significant wave height RMSE at Banks Peninsula.
Mean period comparisons between the forecast and hindcast models show similar results. At Baring Head mean bias, RMSE
and correlation coefficient were -0.31 s, 0.78 and 1.36 s respectively in the SWAN hindcast and -0.34 s, 0.75 and 1.63 s in the
forecast. Improved statistics from both models were found for Banks Peninsula in comparison to Baring Head. Mean period
bias, correlation coefficient and RMSE were 0.09 s, 0.73 and 0.89 in the hindcast and -0.11, 0.89 and 0.65 s in the forecast
(NZWAVE-HR).

Recent developments in WW3 include the implementation of the spherical multiple-cell (SMC) grid system (Li, 2022).
This approach allows for adjustable spatial resolutions within a single model domain, offering the potential for more precise
simulations in coastal regions. It also reduces model resolution in less complex regions which saves computational time.
Therefore, continued validation efforts are crucial as they help identify these specific regions where high-resolution grids are
most beneficial, thereby enhancing our forecasting capabilities. Moreover, investigations towards a 2-way coupled wave-current
forecasting system (e.g. Couvelard et al., 2019; Fragkou et al., 2023) should be conducted.

*Code availability.* WAVEWATCH III is widely known in the modelling community. The code used in this work and its documentation can
be found at https://doi.org/10.5281/zenodo.13867349.



*Author contributions.* R.S. ran extra wave forecast experiments, analysed the results and wrote the manuscript. R.G. set up the wave forecast system. E.L. generated the water level and currents forecast. S.M. is part of the team developing and operating the atmospheric forecast. C.B. analyses and manages wave buoy measurements. G.R. develops the new water level and currents forecast system. C.R. provided support during project execution. R.S., R.G., E.L., S.M., C.B., G.R., and C.R. discussed the results and reviewed the manuscript.

*Competing interests.* No competing interests are present.

*Acknowledgements.* This research was funded by the National Institute of Water and Atmospheric Research (NIWA / Taihoro Nukurangi) Strategic Scientific Investment Fund for "Hazards Exposure and Vulnerability". We thank everyone involved in data collection and curation at NIWA and support from the Aotearoa New Zealand eScience Infrastructure (NeSI).



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
