# Peer review of "Wave forecast investigations on downscaling, source terms, and tides for Aotearoa New Zealand"

_Geoscientific Model Development, 2024_

## Author Response (AR1)

*Dear Editor and Reviewers,*

*We appreciate the invitation to re-submit a revised manuscript incorporating the reviewers' comments. We have revised the paper and corrected the manuscript in response to the thoughtful suggestions received. We are grateful for the comments that we believe have improved the quality of the manuscript. Below, we detail our responses to reviewers' suggestions, addressing the concerns raised.*

*Kind regards,*

*The authors*

**Response to Reviewer 1**

**Reviewer 1 summary:** In the paper 'Wave forecast investigations on downscaling, source terms, and tides for Aotearoa New Zealand' the authors show the results of a set of nested wave models. The results are compared with data from two buoys characterized by different tidal conditions; moreover, the authors discuss the difference in the significant wave height pattern against satellite data. The manuscript is well organized and can help in the setup of new wave simulations.

The performances of the models presented are good. The authors show that the increases in the quality of the results with the model resolution depend on the site conditions. However, some conclusions on the improvement obtained by considering tide in the wave model are based on differences in statistical parameters too small. I think that the authors should take this problem into consideration and underline it in the abstract and in the conclusions. For example, buoy data at Baring Head present peak direction changes of about 80° in a cycle while results from the model experiment including tide show very little modulations both in the wave height and direction. Among the hypotheses presented by the authors to explain the limited impact of tides, there is an underestimation of the current computed by RICOM with respect to that from TELEMAC. It should be interesting to perform a short simulation (just for the period shown in Figure 13) to check the impact of using this alternative forcing.

*Authors' response: Thanks for your comment. We have reviewed the manuscript and modified segments about the inclusion of tides and significant impact on forecast skill. We have included statistics of significant wave height for the short period analysed in Figures 11 and 13 (now Figures 9 and 11). The difference between skill with and without tidal forcing is small and we have modified sections of the manuscript accordingly. We also agree that a comparison between wave forecasts forced by RiCOM and TELEMAC would help us better understand the influence of tides on the wave variability at Baring Head station. However, running a wave simulation forced by TELEMAC is currently beyond achievable scope at the moment due to several changes in the forecast system and institutional supercomputer capabilities. We are working on transitioning the tide and storm surge forecast from RiCOM to TELEMAC. In the*

*near future, we will be able to run a hindcast for 2021 and compare the results between the simulations which we would like to present these results in a future manuscript / submission.*

*To better describe the role of tidal currents on the wave direction, we have included a relation between relative vorticity of the velocity field and wave group velocity proposed by Dysthe (2001, https://doi.org/10.1017/S0022112001005237). Dysthe (2001) describes that when ocean surface waves cross an area with variable current, refraction takes place and the curvature of the wave ray is given by the local flow relative vorticity divided by the wave group velocity. Using this relation, one expects positive relative vorticity to refract waves (or bend wave rays) in one direction and negative relative vorticity to shift waves in the opposite direction given similar wave group velocities. Therefore, we have modified Figures 12 and 14 (now Figures 10 and 12) to show relative vorticity instead of current magnitude which highlight the role of tidal currents on the wave direction variability. We also have modified Section 3.5 "Tidal influence on wave height, period and direction" to describe this process.*

**Specific comments:**

**Line 7-9 The statement: 'Source term enhancements using default ST6 parameters generally improved Hs predictions on the west coast but worsened them on the east, indicating a geographical dependency in model performance." should be postponed, as it seems that the following statement is connected to the previous one.**

*Authors' response: Thanks for that. We have modified that fragment of the Abstract accordingly. Now it reads: "This variability was also evident in the Tm01 predictions, with notable improvements in bias reduction through model downscaling, particularly at Baring Head. Using default source term 6 (ST6) parameters generally improved Hs predictions on the west coast but worsened them on the east, indicating a geographical dependency in model performance."*

**Line 10-11 'Tidal influences were significant, especially at Baring Head, where they enhanced the forecast accuracy of wave height and direction' as I have previously observed I believe this statement cannot be proven by the statistical parameters.**

*Authors' response: We agree with the reviewer. We have modified that fragment to: "Tidal forcing had a small impact on the overall forecast skill and its impact was mostly noticed at Baring Head, where tides force large variability. However, the tidally-driven wave model showed smaller 12-h variability compared to observations."*

**Line 18-20 'Wave climate studies are usually derived from longterm hindcasts or reanalyses. Wave variability can also be described in context of extreme (Simmonds and Keay, 2000) and ambient wave climates (Mortlock and Goodwin, 2015).' I think these statements should be rewritten or eliminated as they are too generic and not connected with the subject of the paper.**

*Authors' response: We agree. We have removed that fragment from the Introduction.*

**Line 45 – In general, I think that it could be better to sign every place cited on the map (i.e. Cook Strait or Hauraki Gulf, Banks Peninsula and Baring Head on the map in Figure 1, Steep Head, and Te Whanganui-a-Tara/Wellington on Figure 2).**

*Authors' response:* We have included the aforementioned locations in Figure 1 and Figure 2 as suggested with the exception of Steep Head. We stopped mentioning Steep Head and only used Banks Peninsula as the general location.

**Line 76-77 'The EcoConnect platform, comprising data ingestion, numerical modelling applications for a variety of natural hazards and forecast data delivery.' I think this phrase is not complete.**

*Authors' response:* We agree. We have modified that text fragment to: "The EcoConnect platform ingests data and runs numerical models to deliver forecasts for a variety of natural hazards."

**Line 82-83 Could you please specify more clearly if the model domains of the atmospheric and wave models coincide?**

*Authors' response:* Yes, they coincide. We have added a sentence at the end of the paragraph to clarify that: "The three domains of the wave models cover the same area as UKMO (global), NZLAM and NZCSM atmospheric models -- more details are provided in the next section."

**I think it could be useful, due to the complexity of the model configuration, add a table with the main details of the wave models, containing for example: atmospheric model forcing, area covered and simulation length, assimilation...**

*Authors' response:* We have added Table 1 in Section 2.6 which shows a summary of all wave forecast experiments.

**Line 260 – It is not clear why you use the wording 'historical wave forecasts'.**

*Authors' response:* We agree with the reviewer. The word "historical" was removed.

**In Figure 10 it seems that the total percentage computed over all the directions doesn't reach 100 in the directional histogram for observations. Could you check?**

*Authors' response:* I rechecked it and Figure 10 (now Figure 8) is accurate. Explanation: the smallest and largest circles correspond to 13% and 65% of occurrences. Peak wave directions from S and SSE (~26%) correspond to ~52% in total. WNW wave directions had 13% of occurrence and four other directions had percentages of occurrence close to 7%. This gives a total of ~93% (26%*2 + 13% +4*7%). The remaining 7% is related to smaller occurrences not visible from the WSW direction and NE quadrant.

**Line 379** – The timeseries of significant wave height and mean period of the NZWAVE-HR simulation show the 12-hour periodicity, however they are very close to those of the NZWAVE-HR-NOTIDES simulation. The authors say: 'which highlights the importance of including tides as forcing'. I think this is a too strong statement to do just looking at the Figure, at least the correlation with data could be computed.

*Authors' response:* We agree with the reviewer. We have computed statistical metrics (RMSE and correlation coefficient) and included them in Figures 9 and 11. We have also modified that text fragment to tone down our argument. Now that part reads: "In this study, the forecast model without tidal and storm surge forcing (NZWAVE-HR-NOTIDES) does not show any 12-h variability in significant wave height and mean period (Tm01) and showed slightly lower skill on forecasting Hs compared to NZWAVE-HR (Fig. 9 and 11)."

**Line 464-470** I think that the authors should change the statement : 'This simulation reduced …' as the difference in the statistical parameters is extremely low. In the same way they should avoid saying: 'but tides didn't show large impact on the wave forecast evaluation statistics…' when discussing results for Bank Peninsula as they have not shown 'large impact' even for the other buoy.

*Authors' response:* We agree with the reviewer. We modified that sentence to: "The improvement in skill is, however, small when looking at the whole-year statistics. RMSE of significant wave height only reduced from 0.27 to 0.26 m and a small increase in the correlation coefficient from 0.94 (NZWAVE-HR-NOTIDES) to 0.95 (NZWAVE-HR) was observed."

We also have modified the fragment about Banks Peninsula and tides to: "At Banks Peninsula, a similar wave-current interaction process occurs but the inclusion of tides in the simulation didn't show any impact on the wave forecast skill."

**Typos and minor:**

**Admas-Bashforth scheme => Adams-Bashforth**

**Line 276 – eliminate 'in these regions'**

**Line 379 – the number of the Figure is missing**

**Line 382 - red rectangles in Figure 11 are not 'Around midnight on the 21st and 22nd of**

**Oct 2021' but around midday and midnight on the 21st of Oct 2021.**

**In the caption of Figure 13 is written 'peak wave period' but it should be 'peak direction'.**

**Lin 416 – 419 please check the description as it seems that period should be changed in peak.**

**Line 463 – 'generates amplitudes I think should be changed in 'generates changes in amplitudes'**

**Line 483 – 486 check the order of the statistics named (mean bias, RMSE and correlation' should 'mean bias correlation and RMSE')**

*Authors' response:* Thanks very much for the corrections. We've modified the manuscript accordingly but the suggestion relative to Line 463. We modified "generates" to "shows".

**Response to Reviewer 2**

**Reviewer 2 summary:** This paper presents the results of a wave forecast system for New Zealand. The sensitivity to the variation of resolution, forcing and tide is considered. The paper is written in good English, and the results are well presented. I have some minor comments, reported below:

*Authors' response:* Thanks for your comments. We have modified the manuscript accordingly.

**Minor comments:**

**- People outside New Zealand are not aware of Maori's names. Moreover, this is used only in some locations and in different ways (with a space or a bar /). This is a bit misleading, I suggest putting one of the two names in parenthesis and specifying it with a short sentence in the Introduction.**

*Authors' response:* We have modified the manuscript accordingly by placing the name of locations in Te Reo Māori between parentheses. We added a sentence at the end of the Introduction to explain that: "In this research article, we acknowledge the existence of names of geographical locations in Te Reo Māori, and we provide them between parentheses. An exception to that is Aotearoa New Zealand, which is widely used in that form."

**- The presentation of the results is a bit too long, with too many figures, scatter plots in particular. Reduce them, maybe you can add a table with some statistics.**

*Authors' response:* Thanks for your comment. We have removed two scatter plots comparing peak wave period between models and observations. We have substituted those Figures by Table 1 which shows the same statistics in a summarised way.

**- Since the wind forces the waves, I would like to see some statistics on its quality.**

*Authors' response:* Thanks for your comment. At the moment, we don't have publications on the newer configurations of the atmospheric models. The most recent results show that simulated winds had a relative error between 2% and 15% during a strong southerly event in Wellington region (Yang et al., 2017, [https://doi.org/10.1175/MWR-D-16-0159.1](https://doi.org/10.1175/MWR-D-16-0159.1)). We have added sentences at the end of Section 2.3 "Atmospheric forcing" providing information on this matter.

**- The tidal interaction is the more physically interesting thing, I suggest expanding this part. Maybe you can add some spectral analysis.**

*Authors' response:* Thanks for your comment. Table 2 in the manuscript shows the effect of the amplitude of M2 tidal constituent on significant wave height and peak direction. M2 causes the largest tidally-driven wave variability. We refrain from including more results in the manuscript which is already lengthy, hence we needed to substitute two figures by one table. We would like to provide a deeper look into the tidal and other variabilities using spectral analysis in another manuscript. This second manuscript would also test newer ocean current and sea level forcing as suggested by another reviewer.

**- p1r7: Define ST6.**

*Authors' response:* Thanks. We defined ST6 as source term 6 in the Abstract.

**- p2r43: remove "refer to".**

*Authors' response:* Removed.

**- Section 2.2.1: Add some citations on the models.**

*Authors' response:* We have included citations on Wave Watch 3 (WW3) and GLOBALWAVE. Wave Watch 3 is mainly cited in Section 2.2. NZWAVE and NZWAVE-HR are new WW3 implementations that are being analysed for the first time in the current manuscript.

**- p8r193-198: I don't understand the use of this model, specify better.**

*Authors' response:* We have included more description about TELEMAC configuration for tides and storm-surge simulation around Aotearoa New Zealand.

**- Tm01, Tm02: explain better the meaning of both these quantities the first time you use one of them.**

*Authors' response:* We have included Tm01's definition (first order mean period) in the abstract. We did the same for Tm02 (second order mean period) in the "Methods". We also include a sentence in the "Methods" to describe these quantities: "These mean periods represent ratios in regard to the zeroth spectral moment and provide different estimates of the average wave period depending on the spectrum's shape."

**- p21r379: Fig. reference is missing.**

*Authors' response:* Thanks. We fixed that.

**- Fig. 11-13: Use different colours and symbols, they are difficult to read.**

*Authors' response: Thanks for raising this issue. We have modified the colour of results from NZWAVE-HR-NOTIDES to magenta in Figures 11 and 13 (now Figures 9 and 11). We have also modified Figures 12 and 14 (now Figures 10 and 12) to show relative vorticity instead of current magnitude. The importance of relative vorticity on the wave direction is shown by Dysthe (2001, https://doi.org/10.1017/S0022112001005237). Dysthe (2001) described that when waves cross an area with variable current, refraction takes place and the curvature of the wave ray (change in wave direction) is given by the local flow relative vorticity divided by the wave group velocity. Using this relation, one expects positive relative vorticity to refract waves (or bend wave rays) in one direction and negative relative vorticity to shift waves in the opposite direction. Section 3.5 "Tidal influence on wave height, period and direction" have been modified accordingly.*